# Near-Interface Defects in Graphene/H-BN In-Plane Heterostructures: Insights into the Interfacial Thermal Transport

**DOI:** 10.3390/nano12071044

**Published:** 2022-03-22

**Authors:** Nana Zhang, Baoming Zhou, Dongbo Li, Dongfeng Qi, Yongling Wu, Hongyu Zheng, Bing Yang

**Affiliations:** 1Centre for Advanced Laser Manufacturing (CALM), School of Mechanical Engineering, Shandong University of Technology, Zibo 255000, China; zhangnn01@sdut.edu.cn (N.Z.); zbm17854336158@163.com (B.Z.); qidongfeng@sdut.edu.cn (D.Q.); ylwu06@sdut.edu.cn (Y.W.); 2Laboratory of Advanced Design, Manufacturing & Reliability for MEMS/NEMS/OEDS, School of Mechanical Engineering, Jiangsu University, Zhenjiang 212013, China; lidongbo1994@163.com

**Keywords:** near-interface defects, interfacial thermal conductance, graphene, H-BN

## Abstract

Based on nonequilibrium molecular dynamics (NEMD) and nonequilibrium Green’s function simulations, the interfacial thermal conductance (ITC) of graphene/h-BN in-plane heterostructures with near-interface defects (monovacancy defects, 585 and f5f7 double-vacancy defects) is studied. Compared to pristine graphene/h-BN, all near-interface defects reduce the ITC of graphene/h-BN. However, differences in defective structures and the wrinkles induced by the defects cause significant discrepancies in heat transfer for defective graphene/h-BN. The stronger phonon scattering and phonon localization caused by the wider cross-section in defects and the larger wrinkles result in the double-vacancy defects having stronger energy hindrance effects than the monovacancy defects. In addition, the approximate cross-sections and wrinkles induced by the 585 and f5f7 double-vacancy defects provide approximate heat hindrance capability. The phonon transmission and vibrational density of states (VDOS) further confirm the above results. The double-vacancy defects in the near-interface region have lower low-frequency phonon transmission and VDOS values than the monovacancy defects, while the 585 and f5f7 double-vacancy defects have similar low-frequency phonon transmission and VDOS values at the near-interface region. This study provides physical insight into the thermal transport mechanisms in graphene/h-BN in-plane heterostructures with near-interface defects and provides design guidelines for related devices.

## 1. Introduction

As typical two-dimensional (2D) materials, graphene and h-BN have outstanding physical and chemical properties, as well as potential application prospects [1,2,3,4]. However, the zero-band gap of graphene greatly limits its application in semiconductor electronic devices. In order to solve this problem, heterogeneous interfaces formed by 2D nitrides and graphene can be used to regulate the electronic properties, such as graphene/g-AlN vertical heterogeneous interfaces [5] and graphene/h-BN in-plane heterogeneous interfaces [6], turning them into promising 2D electronic and photoelectrical materials. H-BN is an insulator with a 5.9 eV bandgap [7,8]. Graphene and h-BN have similar atomic configurations and lattice parameters, and the lattice mismatch between them is less than 2% [9,10,11]. The formation of graphene/h-BN in-plane heterostructures not only opens the band gap of graphene but also has the effect of modulating the intrinsic properties of graphene [12,13]. Ci et al. [6] prepared a graphene/h-BN in-plane heterostructure on copper foil for the first time via the chemical vapor precipitation method, measuring a bandgap value of 18 meV. The results of the previous theoretical predictions and experimental measurements show that the graphene/h-BN in-plane heterostructure has novel electrical properties and unique heat transfer characteristics [14,15,16]. The graphene/h-BN in-plane heterostructure with a zigzag interface exhibits semi-metallic characteristics, while the heterostructure with an armchair interface has semiconductor characteristics [17,18]. In addition to the electrical properties, semiconductor electronic devices also require excellent heat transport capacity. The high-quality, exfoliated suspended graphene measured using the Raman optothermal methods possessed high thermal conductivity up to ∼2000–5000 W/mK near room temperature [19]. Additionally, the thermal conductivity values for h-BN range from 1700 to 2000 W/mK [20]. Due to the fact that graphene and h-BN are excellent phononic heat conductors, they and the heterostructures formed by them have also become effective candidates for the thermal management of semiconductor electronic devices. Additionally, the graphene/h-BN in-plane heterostructure also has a thermal rectification effect due to the differences in the materials at both ends of the interface [21]. The interface has a crucial influence on the thermal properties of the graphene/h-BN in-plane heterostructure.

The vertical graphene/h-BN van der Waals heterogeneous interface is formed by weak van der Waals forces, and the heat transport at the interface is greatly restricted [22]. The graphene/h-BN in-plane heterogeneous interface is bonded together through chemical bonds, which greatly improves the heat transferability at the interface. Although there are many experimental studies devoted to the preparation of graphene/h-BN in-plane heterostructures [2,3,7], defects will inevitably appear in graphene during the growth of h-BN due to its low chemical stability [9]. It is very important to study the effects of defects on the properties of the in-plane heterogeneous interface of graphene/h-BN, especially the effect on the heat transfer performance. Li et al. [23] investigated the effects of monovacancy defects and Stone–Wales (SW) defects on the thermal conductivity of graphene/h-BN in-plane heterogeneous structures using the nonequilibrium molecular dynamics (NEMD) method, and they found that the interfacial thermal conductance (ITC) decreased linearly with increasing concentrations of monovacancy defects. However, as the concentration of SW defects increases, the ITC decreases first and then tends to stabilize. The reason for the decrease in ITC is that the existence of defects will cause stress and ripple effects, reducing the overlap of phonons between graphene and h-BN. Wu et al. [24] studied the effects of topological defects on the ITC of graphene/h-BN in-plane heterostructures, and the results showed that topological defects can improve the heat transport capacity of the interface within a certain defect concentration. This shows that it is feasible to change the ITC of the graphene/h-BN in-plane heterostructure through defect engineering. In addition to studying the thermal conductivity of monolayer graphene/h-BN in-plane heterostructures, Fan et al. [8] investigated the effects of defects and interlayer sp^3^ bonds on the thermal transport of bilayer graphene/h-BN in-plane heterostructures explored through molecular dynamics (MD) simulations. When the concentration of interlayer sp^3^ bonds is lower than 2%, the ITC is reduced, which indicates that the sp^3^ bond has a defect effect. The coupling of sp^3^ bonds and defects can result in a defect amplifier effect. It is of great significance to fully analyze and understand the influence of defects on the ITC of graphene/h-BN in-plane heterogeneous interface.

Herein, the effects of monovacancy defects and 585 and f5f7 (5555-6-7777) double-vacancy defects on the ITC of graphene/h-BN in-plane heterostructures are investigated by NEMD and nonequilibrium Green’s function simulations. Three different kinds of defects are regularly distributed in the interface area. The defective structures, interfacial wrinkles, phonon transmission and vibrational density of states (VDOS) are used to analyze the mechanism of interface heat transfer. Our research aims to provide a theoretical guidance for the thermal management of graphene/h-BN in-plane heterostructure-based electronic devices.

## 2. Materials and Methods

The interfaces of the graphene/h-BN in-plane heterostructures include zigzag interfaces and armchair interfaces. Furthermore, zigzag interfaces can be divided into two bonding types, namely boron atoms are bonded with carbon atoms at the interface and nitrogen atoms are linked with carbon atoms at the interface. Since the zigzag interfaces with either boron–carbon or nitrogen–carbon links have smaller formation energy values than that of the armchair interface, the former is more likely formed. Researchers have mostly focused on the heat transfer of graphene/h-BN in-plane heterostructures with zigzag interfaces [24,25,26]. However, the graphene/h-BN in-plane heterostructures with armchair interfaces have also been prepared in experiments [15], and they show a higher bandgap as compared to the pristine nanoribbons [17,27]. The opening of the bandgap makes them potentially important for applications in semiconductor devices. Combined with the heat transfer requirements of semiconductor devices, insight into their thermal transport properties is equally necessary. Thus, the heat transfer of a pristine and defective graphene/h-BN in-plane heterostructure with armchair interface is studied systematically. The schematic of the atomistic configuration of the pristine graphene/h-BN in-plane heterostructure with the armchair interface is shown in Figure 1. The lattice constants of graphene and h-BN are $a_{c}=2.46$ Å and $a_{h-BN}=2.52$ Å, respectively. The corresponding lattice mismatch strain is $\chi =\frac{a_{h-BN}-a_{C}}{a_{C}}=2.44\%$. It is noteworthy that the ITC exhibits a strong size dependence when the simulated size is smaller than the phonon mean free path in nanosystems. In order to only observe the effect of the changes in defects on interfacial heat transport, the single variable method is adopted. Therefore, the sizes of the model are fixed. In all simulated systems, the lengths of graphene and h-BN are $L_{C}=$ 172.23 Å and $L_{h-BN}=$ 173.40 Å, respectively. The widths of both graphene and h-BN are $W_{C}=W_{h-BN}=$ 127.80 Å. A model description of defective graphene/h-BN is provided in Section 3.1.

Here, the large-scale molecular dynamics simulations (LAMMPS) package [28] is used to calculate the ITC. The Tersoff potential [29,30], whose parameters originate from the literature [31], is used to study the thermal properties of graphene/h-BN in-plane heterostructures. In a previous study, Tersoff potential values were developed for all possible element pairs at the interface of graphene and h-BN. Specifically, the researchers generated the data needed for the interfaces by using density functional theory (DFT) energetics to condition empirical Tersoff potential values. Finally, they found that MD energy levels were in good agreement with DFT results, and the errors in the calculated equilibrium B–C and N–C bond lengths for all structures were no larger than 1.5%. Other studies [24,31,32] performed accurate calculations of the thermal conductivity of hybrid nanostructures formed by graphene and h-BN. Here, the shrink-wrapped boundary conditions are used in all directions. The shrink-wrapped boundary conditions are nonperiodic, which could lead to boundary scattering for the models used in this work. In the process, the *x*-axis direction is set as the heat flow direction with heat flowing from the graphene to the h-BN. The y-axis is parallel to the graphene/h-BN interface. The atoms of two hexagon widths at the ends of the heat flow direction are fixed to prevent atomic drift. After minimizing the energy of the system by iteratively adjusting the atomic coordinates, the system relaxes in the NVT ensemble at a temperature of 300 K and with a time step of 0.5 fs for 1 ns. When the system temperature reaches 300 K and fluctuates for an extended time, this indicates that the system has reached the equilibrium state. After this, the system is transferred to the NVE ensemble and continues to relax for 0.5 ns. Then, the NEMD simulation is performed, and a heat source and heat sink with five hexagonal widths at each end of the heat flow direction are introduced using the Langevin thermostat [33]. It should be noted that the sizes of the heat source and heat sink are fixed in all simulations in order to only observe the effects of the changes of the defects on interfacial heat transport. The temperatures of the heat source and heat sink are 270 K and 330 K, respectively. Within 3 ns, as shown in Figure 1a, the system forms a stable spatial temperature profile. Meanwhile, as shown in Figure 1b, the cumulative energy of the heat source located in the graphene domain and the heat sink in the h-BN domain with simulation time is assessed. The data show that the sum of the increased and decreased energy is 0, demonstrating that the system is in a state of energy conservation. Before calculating the ITC, the unstable temperature and energy data in the first 0.5 ns are removed first, and the average temperature gradient T(x) and energy exchange rate Q(t) are obtained using temperature and energy data in the last 2.5 ns. Figure 1 shows that the temperature profiles at the heat source and the heat sink are nonlinear, originating from the phonon scattering that occurs in the heat source and heat sink [34]. After removing the heat source and heat sink regions, both graphene and h-BN have linear temperature profiles with a certain temperature jump ∆*T* at the interface. The ITC can be obtained from the ratio of the heat flow density *J(t)* and the temperature jump ∆*T* as follows:(1)$$G=\frac{J\left(t\right)}{\Delta T}$$
(2)$$J\left(t\right)=\frac{Q\left(t\right)}{S}$$
(3)$$Q\left(t\right)=\frac{\sum_{N_{swap}}\frac{1}{2}\left(m_{1}v_{hot}^{2}-m_{2}v_{cold}^{2}\right)}{N_{swap}\times t_{swap}}$$
where *S* is the cross-sectional area at the interface, the thickness of which is 0.34 nm [26,35]; $m_{1}$ and $m_{2}$ are the masses of atoms in the heat source and the heat sink, respectively; $v_{hot}$ and $v_{cold}$ represent the minimum atomic velocity in the heat source and the maximum atomic velocity in the heat sink, respectively; $N_{swap}$ refers to the number of particle exchange rates between the heat sink and the heat source in each time step; $t_{swap}$ is the exchange speed time interval.

## 3. Results and Discussion

### 3.1. The ITC Results of Pristine Graphene/h-BN and Graphene/h-BN with Near-Interface Defects

Many graphene-like materials, such as 2D graphene/h-BN, 2D nitride and 2D oxide, can be grown from the surface or interface of a substrate using chemical vapor deposition (CVD) [36,37]. However, defects can inevitably be introduced in the growth process. During the growth of h-BN starting from CVD graphene edges on copper, defects may form in graphene domains due to its relatively low chemical stability [9]. These defects significantly affect the thermal transport properties of graphene/h-BN in-plane heterostructures. Based on transmission electron microscope (TEM) and scanning electron microscope (SEM) observations at the atomic resolution level, graphene defects are classified into two categories. One group contains intrinsic defects, which consist of carbon atoms with non-sp^2^ orbital hybridization in graphene, with configurations of nonhexatomic ring, point or line domain voids. The other group contains externally introduced defects, which are caused by noncarbon atoms being covalently bound to the carbon atoms in graphene. Due to the different atomic types, the external atomic defects strongly affect the charge distribution and properties of graphene.

The intrinsic defects include the SW defect, monovacancy defect, double-vacancy defects (such as 585 and f5f7) and line defects. They have their own unique structures and formation energy levels, as well as thermal stability at room temperature [38,39,40]. The SW defect involves the transformation of four hexagons into two pentagons and two heptagons by rotating one of the C–C bonds by 90°, and its formation energy is about 5 eV. In the monovacancy defect, the loss of one carbon atom causes the breakage of three covalent bonds originally attached to it, forming three dangling bonds. In order to minimize the overall energy, structural rearrangement occurs in the region of lost carbon atoms of graphene under the Jahn–Teller effect [41]. Eventually, two dangling bonds join each other with one dangling bond remaining, meanwhile the structure of the region is adjusted and the level protrudes [42]. The formation energy range for this defect is approximately 7.3–7.5 eV. The 585 double-vacancy defect consists of a pair of pentagons and one octagon, which is formed by the loss of the carbon atom with the dangling bond in the monovacancy defect, the formation energy range of which is about 7.2–7.9 eV. The f5f7 double-vacancy defect consists of four pentagons and four heptagons and surrounds a hexagon in the middle, the formation energy of which is about 7 eV. The line defects are tilt boundaries separating two domains of different lattice orientations, with the tilt axis positioned normal to the plane. They can be considered reconstructed point defect lines with or without dangling bonds. These defects are all observable under TEM and STM. Additionally, once these defects are formed under nonequilibrium conditions, the barriers in the range of 5–7.9 eV for the reverse transformation should maintain their stability at room temperature [38]. It is noteworthy that defects must be constructed based on actual conditions. Monovacancy defects in graphene and h-BN have different configurations. The monovacancy defect in graphene undergoes structural rearrangement [38]. However, no structural rearrangement occurs in monovacancy defect in h-BN in the absence of the boron atom, as shown via electron micrography [43]. Moreover, theoretical calculations indicate that the absence of the boron atom increases the charge number between the two nitrogen atoms, and the increased Coulomb repulsive force causes the distance between nitrogen atoms to increase [44]. In addition, simulations show that after replacing the nitrogen atoms with dangling bonds at the defects with carbon atoms, the carbon atoms also form carbon–carbon bonds similar to the carbon atoms with dangling bonds in graphene and undergo structural reconfiguration [44]. This indicates that divergence in interactions between atoms with dangling bonds can lead to structural differences; thus, models must be constructed on realistically. Because the defects in graphene/h-BN may appear in the near-interface area of the graphene region [9], combined with the structural characteristics of the defects in graphene [38], here we construct monovacancy defects and 585 and f5f7 double-vacancy defects, as shown in Figure 2. It should be noted that these three kinds of defects can exist stably at the simulated temperature (300 K).

Here, the ITC values of pristine graphene/h-BN and graphene/h-BN with different defect types and numbers of defects at room temperature (300 K) are calculated. The results are summarized in Figure 2. The ITC of pristine graphene/h-BN is 12.76 ± 0.50 GW/m^2^K, which is in good agreement with the previous molecular dynamics simulation results [25,26,45]. Figure 2 also shows that the ITC for defective graphene/h-BN is smaller than that of pristine graphene/h-BN, and all ITC values decrease with the increasing number of defects. The maximum decreases are 25.05% for monovacancy defects, 34.56% for 585 double-vacancy defects and 35.70% for f5f7 double-vacancy defects. This phenomenon is similar to the effect of defects observed experimentally on the thermal transport of graphene. Renteria et al. [46] found that a decrease in defect concentration can increase the thermal conductivity of free-standing reduced graphene oxide films. Malekpour et al. [47] studied the relationship between the thermal conductivity of suspended graphene and the defect density. Here, the results showed that when the defect density increases from 2.0 × 10^10^ cm^−2^ to 1.8 × 10^11^ cm^−2^, the thermal conductivity changes from ∼(1.8 ± 0.2) × 10^3^ W/mK to ∼(4.0 ± 0.2) × 10^2^ W/mK. In addition, it is found that the reduction in ITC of graphene/h-BN with double-vacancy defects is greater than that with monovacancy defects, while the ITC of graphene/h-BN with either 585 double-vacancy defects or f5f7 double-vacancy defects differ little with different numbers of defects. In order to clarify the mechanism behind the changes in ITC, the defects and induced wrinkles in models as well as the corresponding phonon transmission and VDOS are further analyzed.

### 3.2. Near-Interface Defects and Induced Wrinkles

The decrease in ITC arises from the weakening of the phonon transport capacity caused by the near-interface defects and the wrinkles in the near-interface region induced by the near-interface defects.

When phonons hit graphene defects during the transport from the heat source to the heat sink, some phonons can move forward through the defects without changing the transport direction or by changing the direction at a small angle, while others would change the transport direction at a large angle or even move backward, producing scattering. Scattering is the intrinsic mechanism of defects affecting heat transfer characteristics. Our analysis indicates that the size of the cross-section of the defect is positively correlated with the scattering ability. For 2D materials, the cross-section of the defect can be measured by the width of the defect in the direction perpendicular to the transmission path. If the cross-section is larger, more phonons would be scattered at the defects. However, more phonons would pass smoothly on the sides of the defects without being affected by the defect when the cross-section is decreased. Therefore, it can be expected that the larger the cross-section of the defect, the more difficult the heat transfer will be. As the parallel lines show in Figure 2, the cross-sections of the three intrinsic defects after sufficient relaxation are provided. These parallel lines are aligned to the left to facilitate comparative analysis. From this point of view alone, the monovacancy defect has a smaller cross-section compared to the double-vacancy defects. Therefore, the monovacancy defect causes less phonon scattering. In addition, the similar cross-sections of 585 and f5f7 double-vacancy defects cause comparable phonon scattering. Moreover, the bond energy of C–C in graphene is 770 kcal/mol [48], while the heat transfer is closely related to the C–C bond. Compared to pristine graphene/h-BN, the region including defects has fewer C–C bonds. The number of C–C bonds in graphene/h-BN with 585 or f5f7 double-vacancy defects is less than that in graphene/h-BN with monovacancy defects. Thus, the reduction in interaction strength decreases the ability for heat transfer.

Further, the effect of wrinkles in the models on the ITC is analyzed. The wrinkles are caused by the presence of defects leading to regional restructuring, which further causes level protrusions. Similar phenomena have been observed in previous experiments [38]. Research indicates [49] that in contrast to pristine graphene, graphene with wrinkles exhibits relatively low thermal conductivity. This stems from the strong phonon localization leading to the concentration of phonons in the connection region between the crests and troughs of the wrinkles. It also stems from the enhanced phonon scattering due to the wrinkles. As shown in Figure 3, the wrinkles in the near-interface regions of the pristine graphene/h-BN and graphene/h-BN with intrinsic defects after sufficient thermal relaxation are plotted. First, it can be seen from Figure 3a–d that the fluctuations of the wrinkles in the near-interface region intensify with the increase in f5f7 double-vacancy defects, which leads to enhanced phonon scattering and blocked heat transfer in the near-interface region. Therefore, the ITC becomes smaller with the increase in defects. Additionally, comparing Figure 3a,d–f, it can be seen that the presence of the defects causes greater wrinkles than the absence of defects. Thus, when phonons propagate to the near-interface region with defects, the presence of the wrinkles intensifies the phonon scattering, and the wrinkles also cause the localization of phonons, which in turn leads to block phonon transmission and causes a lower ITC in the near-interface region. Moreover, analyzing Figure 3d–f, the wrinkles caused by double-vacancy defects are larger than those induced by monovacancy defects, meaning the former have lower ITC values.

### 3.3. Phonon Transmission

The nonequilibrium Green’s function is an effective method for studying quantum heat transport in nanosystems [40,50]. In this method, the Landauer formula [51] is used to calculate the thermal conductance. Equation (4) shows that the thermal conductance is proportional to the phonon transmission coefficient at the same temperature. Therefore, the phonon transmission coefficient can be used to evaluate the capability for heat transport:(4)$$\sigma =\frac{\hbar^{2}}{2\pi k_{B}T^{2}}\int_{0}^{\infty }dw\frac{w^{2}e^{\frac{\hbar w}{k_{B}T}}}{{\left(e^{\frac{\hbar w}{k_{B}T}}-1\right)}^{2}}T\left(w\right)$$
(5)$$T\left(w\right)=Tr\left[G^{r}\left(w\right)\Gamma_{L}\left(w\right)G^{a}\left(w\right)\Gamma_{R}\left(w\right)\right]$$
where ℏ is the approximate Planck constant, $k_{B}$ refers to the Boltzmann constant, *T* is the system temperature, *w* represents the phonon frequency, $T\left(w\right)$ is the phonon transmission coefficient at the frequency *w* and $\Gamma_{L}\left(w\right)$ and $\Gamma_{R}\left(w\right)$ are the escape rates of phonons with frequency *w* from the left and right contacts, respectively. Here, $\Gamma_{L}\left(w\right)$ corresponds to the bottom contact in Figure 4, $\Gamma_{B}\left(w\right)$ corresponds to the top contact in Figure 4 and $G^{r}\left(w\right)$ and $G^{a}\left(w\right)$ are the Green’s function and the complex conjugate of the heat transfer structure region, respectively.

In order to further illustrate the effects of the defects and wrinkles induced by the defects on phonon transport at different frequencies, phonon transmission values in the near-interface regions of pristine graphene/h-BN and graphene/h-BN with twelve monovacancy or double-vacancy defects at different frequencies are calculated. The basic structure of the graphene/h-BN in-plane heterogeneous interface is constructed according to the computational requirements of the phonon transmission function. Figure 4 shows the basic structure of the graphene/h-BN in-plane heterogeneous interface with twelve f5f7 double-vacancy defects as an example, with the positions of other defects consistent with those of f5f7 double-vacancy defects. Twelve defects are equally spaced along the direction perpendicular to the heat flow, which is consistent with the calculation of the ITC based on the molecular dynamics. The top and bottom ends of Figure 4 are the thermal reservoirs, which correspond to the right and left ends of Equation (5), while the middle is the interfacial structure region for phonon transmission. Before the calculation of the phonon transmission, the structure is relaxed to minimize the energy of the system. Meanwhile, the convergence criteria are 0.0001 eV/Å for the maximum force, 0.01 eV/Å^3^ for the maximum stress and 0.5 Å for the maximum displacement, while the Monkhorst–Pack grid is set to 12 × 12. The potential is the same as that used for the ITC calculation based on molecular dynamics.

The phonon transmission coefficients in the near-interface regions of pristine graphene/h-BN and graphene/h-BN with twelve defects are shown in Figure 5. The phonon transmission shows that the phonon frequencies with heat transfer contributions in all graphene/h-BN samples are mainly in the range of 0–37 THz, consistent with the main heat transfer phonon frequencies (0–40 THz) of graphene [52]. Further, the results show that the phonon transmission coefficients of all defective graphene/h-BN samples are weaker than those of pristine graphene/h-BN in the range of 0–37 THz, indicating that the defects reduce the phonon transmission ability in the near-interface region. Moreover, the phonon transmission coefficients of graphene/h-BN with double-vacancy defects are all lower than those of graphene/h-BN samples with monovacancy defects, demonstrating that the double-vacancy defects are more detrimental to the phonon transport than the monovacancy defects. Finally, the results also show that the phonon transmission coefficients of graphene/h-BN with 585 and f5f7 double-vacancy defects are similar, revealing that the two double-vacancy defects have an approximate phonon transport capacity. This is consistent with the calculated ITC values.

### 3.4. Vibrational Density of States

The differences in ITC values between pristine graphene/h-BN and graphene/h-BN with different defects can be further attributed to the VDOS. The VDOS is obtained from the Fourier transform of the velocity autocorrelation function of all atoms [53]:(6)$$\mathrm{VDOS}(w)=\int_{-\infty }^{+\infty }\langle \frac{1}{N}\overset{N}{\underset{i=1}{\sum }}\vec{v_{i}}{(t}_{0})-\vec{v_{i}}{(t}_{0}{+t)\rangle e}^{-2\pi iwt}dt\;$$
where *N* is the total number of atoms, *w* refers to the phonon frequency and $\vec{\nu_{i}}\left(t\right)$ represents the velocity of the *i*th atom at time *t*. When VDOS(*w*) has a higher value, this indicates that more states are occupied by the phonon with frequency *w*. When VDOS(*w*) is equal to 0, there is no phonon at frequency *w* inside the system.

As shown in Figure 6, the total VDOS, the decomposed in-plane VDOS and the decomposed out-of-plane VDOS for pristine graphene/h-BN are calculated. For all VDOS, the out-of-plane bending (ZA) mode appears in the low-frequency range. The in-plane modes, including in-plane longitudinal acoustic (LA) and transverse acoustic (TA) modes, are concentrated in the high-frequency region, which is consistent with previous studies [23,25].

The above phonon transmission results show that the phonon frequencies with thermal conduction contributions in graphene/h-BN are in the range of 0–37 THz, which is consistent with the phonon frequencies of the out-of-plane VDOS. Therefore, the out-of-plane VDOS results for pristine graphene/h-BN and graphene/h-BN with twelve defects are compared. Figure 7a–c shows that the out-of-plane VDOS results of all the defective graphene/h-BN are weaker than for the pristine graphene/h-BN. Moreover, the out-of-plane VDOS of graphene/h-BN with double-vacancy defects in Figure 7d,e is lower than that of graphene/h-BN with monovacancy defects. This indicates that the defects and wrinkles lead to the increase in phonon scattering, and the double-vacancy defects and their induced wrinkles cause more intense phonon scattering than the monovacancy defects and their induced wrinkles. This is in agreement with the calculated ITC results.

## 4. Conclusions

To conclude, the ITC values were studied for pristine graphene/h-BN and graphene/h-BN with near-interface defects. The graphene/h-BN with near-interface defects had a lower ITC, indicating that it has potential applications in thermoelectric devices. Differences in defective structures and the wrinkles induced by the defects caused significant discrepancies in heat transfer for the defective graphene/h-BN. The ITC values for pristine graphene/h-BN, graphene/h-BN with monovacancy defects and graphene/h-BN with double-vacancy defects decreased sequentially, while the 585 and f5f7 double-vacancy defects had similar heat-blocking ability. Through the analysis of defective cross-sections, wrinkles induced by defects, phonon transmission and VDOS, the underlying mechanism of the reductions in ITC of graphene/h-BN in-plane heterostructures with near-interface defects was confirmed. It was found that the double-vacancy defects have a wider cross-section, larger wrinkles and fewer C–C bonds than the monovacancy defects, causing stronger phonon scattering and phonon localization as a direct result of a reduction in phonon transmission in low-frequency region and a decrease in out-of-plane VDOS. In addition, the 585 and f5f7 double-vacancy defects have similar cross-sections, and the phonon transmission and out-of-plane VDOS in the low-frequency region are similar; thus, they have similar heat-hindering abilities. The results in this paper provide theoretical guidance for the further utilization of graphene/h-BN in-plane heterostructures with near-interface defects.

## Figures and Tables

**Figure 1 nanomaterials-12-01044-f001:**
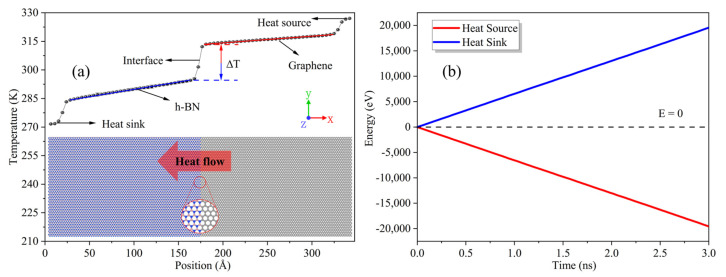
(**a**) At a system temperature of 300 K, the typical steady-state temperature profile of the pristine graphene/h-BN in-plane heterostructure was calculated using the NEMD method. (**b**) Using a Langevin thermostat, the cumulative energy of the heat source located in the graphene domain and the heat sink of the h-BN domain varied with simulation time.

**Figure 2 nanomaterials-12-01044-f002:**
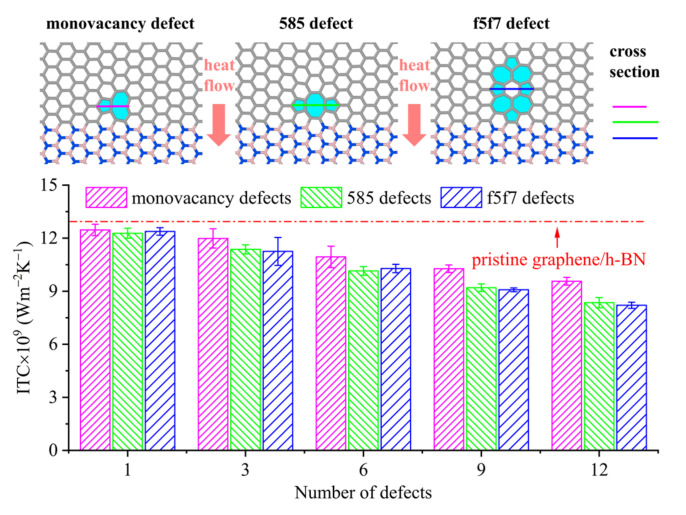
The ITC values of pristine graphene/h-BN and graphene/h-BN with different types and numbers of defects. The inset shows schematic diagrams and cross-sections of the different defects.

**Figure 3 nanomaterials-12-01044-f003:**
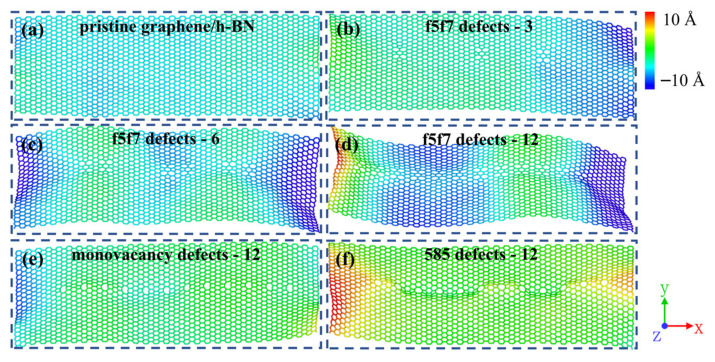
Wrinkles in the near-interface region of pristine graphene/h-BN and graphene/h-BN with intrinsic defects after sufficient relaxation: (**a**) pristine graphene/h-BN; (**b**) graphene/h-BN with three f5f7 double-vacancy defects; (**c**) graphene/h-BN with six f5f7 double-vacancy defects; (**d**) graphene/h-BN with twelve f5f7 double-vacancy defects; (**e**) graphene/h-BN with twelve monovacancy defects; (**f**) graphene/h-BN with twelve 585 double-vacancy defects.

**Figure 4 nanomaterials-12-01044-f004:**
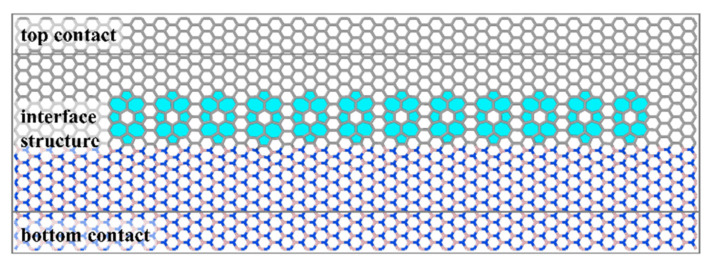
The basic structure of graphene/h-BN in-plane heterogeneous interface with twelve f5f7 double-vacancy defects according to the computational requirements of the phonon transmission function.

**Figure 5 nanomaterials-12-01044-f005:**
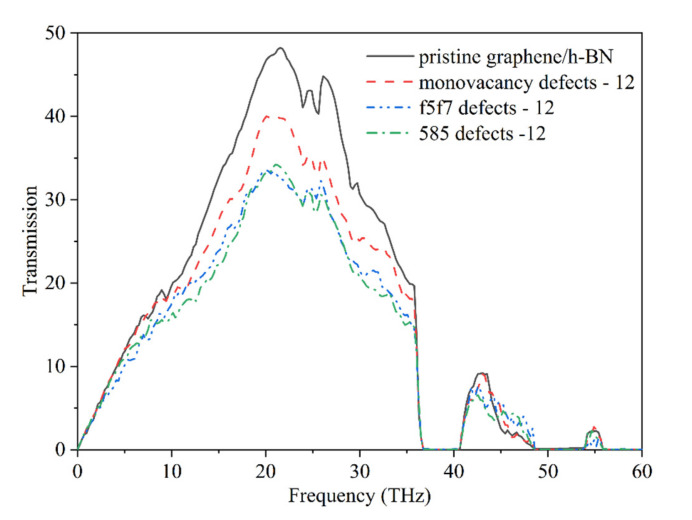
Phonon transmission in the near-interface regions of pristine graphene/h-BN and graphene/h-BN with twelve defects.

**Figure 6 nanomaterials-12-01044-f006:**
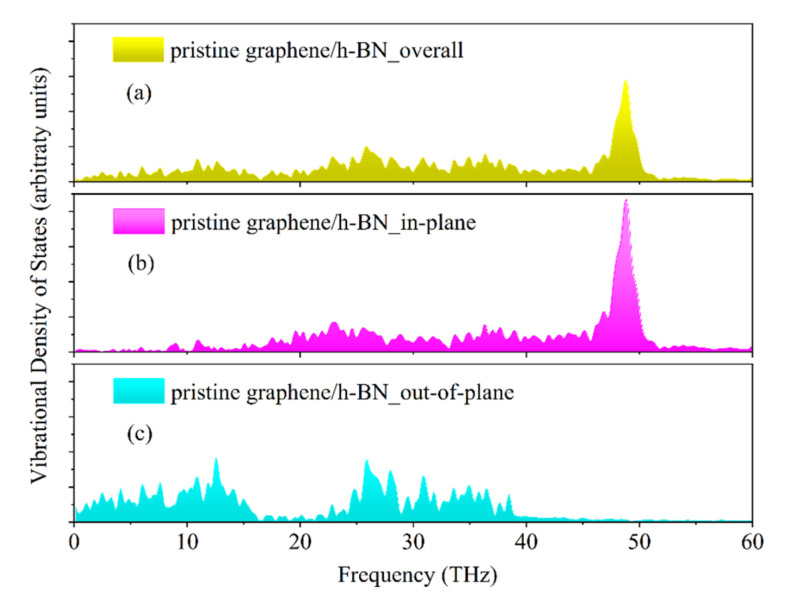
(**a**) Total VDOS, (**b**) in-plane VDOS and (**c**) out-of-plane VDOS in the near-interface region of pristine graphene/h-BN.

**Figure 7 nanomaterials-12-01044-f007:**
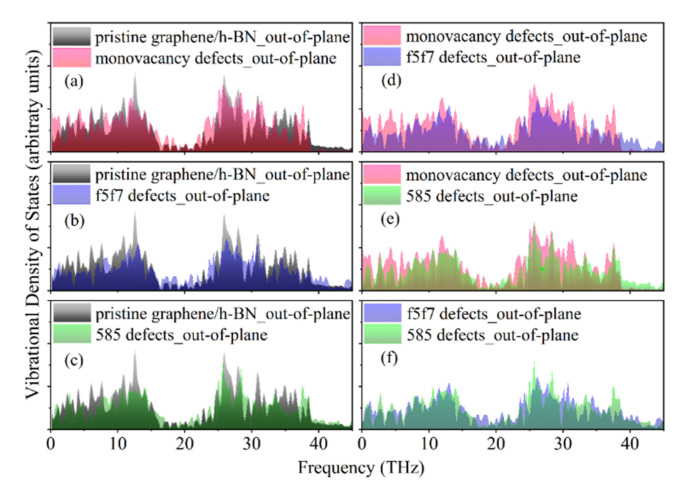
Comparison of the out-of-plane VDOS results in the near-interface regions of pristine graphene/h-BN and defective graphene/h-BN: (**a**) pristine graphene/h-BN and graphene/h-BN with monovacancy defects; (**b**) pristine graphene/h-BN and graphene/h-BN with f5f7 double-vacancy defects; (**c**) pristine graphene/h-BN and graphene/h-BN with 585 double-vacancy defects; (**d**) graphene/h-BN with monovacancy defects and graphene/h-BN with f5f7 double-vacancy defects; (**e**) graphene/h-BN with monovacancy defects and graphene/h-BN with 585 double-vacancy defects; (**f**) graphene/h-BN with f5f7 double-vacancy defects and graphene/h-BN with 585 double-vacancy defects.

## Data Availability

All data are included in this paper.

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
