# Peer review of "Near-Interface Defects in Graphene/H-BN In-Plane Heterostructures: Insights into the Interfacial Thermal Transport"

_nanomaterials, 2022, doi:10.3390/nano12071044_

Round 1

Reviewer 1 Report

Near-interface defects in graphene/h-BN in-plane heterostructures: insights into the interfacial thermal transport

The authors used nonequilibrium molecular dynamics and nonequilibrium Green's function methods to study the interfacial thermal conductance of graphene/h-BN heterostructures with near-interface defects (monovacancy defects, double vacancy, etc.). The results may be useful for experimentalists and thermal management applications. I can recommend the paper for publication after the authors addressed the issues below.

1) The authors study thermal effects in graphene/BN. They should explain early in the introduction why the thermal properties of these materials are important. The main reason is that graphene and hBN are excellent phononic heat conductors. Indicate the intrinsic thermal conductivity of graphene (above 2500 W/mK at RT) and provide relevant references to the review papers on thermal properties ( “Phononics of graphene and related materials,” ACS Nano, 14, pp. 5170-5178, 2020; Y. Fu, et al., “Graphene related materials for thermal management,” 2D Mater., 7, 012001, 2019.)

2) MD simulations often depend on the domain size and the type of boundary conditions used. How the domain size affects the results and, possibly, conclusions, in this work?

3) How interatomic potentials between graphene and h-BN are selected and experimentally justified?

4) The paper needs more connections to the experimental results available in the literature. There have been some studies of the effects of the defects, interfaces etc. The prior studies should be cited and results compared when possible. The relevant literature includes H. Malekpour, et al., “Thermal conductivity of graphene with defects induced by electron beam irradiation,” Nanoscale, 8, 14608–14616, 2016; J. D. Renteria, et al., “Strongly anisotropic thermal conductivity of free-standing reduced graphene oxide films annealed at high temperature,” Adv. Funct. Mater., 25, 4664–4672, 2015; S. Sudhindra, et al., "Noncured graphene thermal interface materials for high-power electronics: minimizing the thermal contact resistance" Nanomaterials, 11, 1699, 2021.

Reviewer 2 Report

This manuscript is dedicated to study of the near-interface defects in graphene/h-BN in-plane heterostructures. The authors focus on the insights and impacts of these defects into the interfacial thermal transport. The work offers a sophisticated and well-planned simulation setup complemented by an adequate and deep aanalysis to achieve better understanding of the structural and functional aspects of near-interface defects in relation to graphene and graphene-like materials such as h-BN (work can have implications for other 2D sheets such as other interfaces between group III nitrides – h-AlN, h-InN, etc)

Thus, the ambitious task in this work covers an array of hot topics of research of interfaces between 2D graphene-like materials and 2D sheets of h-III-nitrides with wide perspectives for applications impacting emerging technologies from energy materials to electronic materials that are currently attracting much research interest.

The authors chose an adequate structure of the manuscript – an excellent point of departure for such a study. Finally, the authors provided a balanced and realistic presentation of their simulation efforts and corresponding results that is of much scientific and practical interest and adds to the vast knowledge in the realm of 2D sheets and their interfaces.

In fact, such work should be seen as long due because the manuscript provides very welcome highly focused study of deffects at interfaces for 2d graphene-like and graphitic materials with plausible usefulness for real-world development and applications in energy materials.

In my opinion, the fine detailing in the present work, the insightful and balanced discussion of the results, as well as the very good figures, permit competent readers to utilize the manuscript as a guidance for future work. Consequently, this manuscript presents an efficient and beneficial basis for promoting and solving next step challenges in this field.

Moreover, the manuscript benefits from a clear motivation and it is an easy and informative read. The manuscript is also excellent in terms of clarity and accuracy of language.

The present manuscript is a significant contribution, this work once published would be quite useful as well as instructive and suggestive in terms of further studies and to a wider readership.

There are some minor issues with this already excellent manuscript that will need to be addressed before becoming suitable for publication, i.e., it can be considered for publication after a minor revision:

1: The authors miss part of bigger picture of 2D graphene-like and graphitic materials for their structure/bonding and electronic properties at the interface. Similar structural aspects as in this work were theoretically addressed for 2D materials such as nitrides, oxides and graphene so these analogies should be mentioned in the present manuscript, e.g., Nanoscale 12 (2020) Pages 19470 – 19476; Applied Surface Science 548 (2021) Article number 149275. Such works are directly supportive to the credibility of the present work.

2: Authors discuss very well aspects of thermal properties (conductivity, transport, etc.) Wherever possible, thermal ranges and, notably, thermal stability of the the defects under question should be discussed in the context of bonding particularities.

3: Again, wherever possible the more concentrated and easy to grasp statistics/presentation of the studies defects (such as pentagons, octagons, Stone Wales defects, etc.) should be made directly explicit to the reader (maybe in a specific paragraph dedicated to statistics of the defects).

4: Spell-check and stylistic revision of the paper are still necessary. Some long sentences, misspellings, etc., still are noticeable throughout the text.
